# Interplay of Cytokines and Chemokines in Aspergillosis

**DOI:** 10.3390/jof10040251

**Published:** 2024-03-27

**Authors:** Jata Shankar, Raman Thakur, Karl V. Clemons, David A. Stevens

**Affiliations:** 1Genomic Laboratory, Department of Biotechnology and Bioinformatics, Jaypee University of Information Technology, Waknaghat Solan 173234, Himachal Pradesh, India; 2Department of Medical Laboratory Science, Lovely Professional University, Jalandhar 144001, Punjab, India; ramanthakurnegwal@gmail.com; 3California Institute for Medical Research, San Jose, CA 95128, USA; kvclemons@centurylink.net (K.V.C.); stevens@stanford.edu (D.A.S.); 4Division of Infectious Diseases and Geographic Medicine, Stanford University Medical School, Stanford, CA 94305, USA

**Keywords:** aspergillosis, cytokines, chemokines, invasive aspergillosis, allergic bronchopulmonary aspergillosis, interleukins, conidia, T cells, reactive oxygen species

## Abstract

Aspergillosis is a fungal infection caused by various species of *Aspergillus*, most notably *A. fumigatus*. This fungus causes a spectrum of diseases, including allergic bronchopulmonary aspergillosis, aspergilloma, chronic pulmonary aspergillosis, and invasive aspergillosis. The clinical manifestations and severity of aspergillosis can vary depending on individual immune status and the specific species of *Aspergillus* involved. The recognition of *Aspergillus* involves pathogen-associated molecular patterns (PAMPs) such as glucan, galactomannan, mannose, and conidial surface proteins. These are recognized by the pathogen recognition receptors present on immune cells such as Toll-like receptors (TLR-1,2,3,4, etc.) and C-type lectins (Dectin-1 and Dectin-2). We discuss the roles of cytokines and pathogen recognition in aspergillosis from both the perspective of human and experimental infection. Several cytokines and chemokines have been implicated in the immune response to *Aspergillus* infection, including interferon-γ (IFN-γ), tumor necrosis factor-α (TNF-α), CCR4, CCR17, and other interleukins. For example, allergic bronchopulmonary aspergillosis (ABPA) is characterized by Th2 and Th9 cell-type immunity and involves interleukin (IL)-4, IL-5, IL-13, and IL-10. In contrast, it has been observed that invasive aspergillosis involves Th1 and Th17 cell-type immunity via IFN-γ, IL-1, IL-6, and IL-17. These cytokines activate various immune cells and stimulate the production of other immune molecules, such as antimicrobial peptides and reactive oxygen species, which aid in the clearance of the fungal pathogen. Moreover, they help to initiate and coordinate the immune response, recruit immune cells to the site of infection, and promote clearance of the fungus. Insight into the host response from both human and animal studies may aid in understanding the immune response in aspergillosis, possibly leading to harnessing the power of cytokines or cytokine (receptor) antagonists and transforming them into precise immunotherapeutic strategies. This could advance personalized medicine.

## 1. Introduction to *Aspergillosis*

The *Aspergillus* genus contains ~250 recognized species [1], of which 40 cause infections, ranging from allergic to invasive aspergillosis [2]. These fungi are commonly found in soil, organic debris, and plants [3,4]. The infection can affect the lungs, sinuses, and sometimes other organs [4,5,6]. In recent years, there has been increasing interest in understanding the role of *Aspergillus* in various respiratory diseases and conditions [7,8]. Aspergillosis can range from mild allergic reactions to severe invasive disease, particularly in individuals with weakened immune systems. *Aspergillus fumigatus* is the most prevalent cause of aspergillosis in immunocompromised patients [5,6,7]. The species that causes invasive aspergillosis most often is *A. fumigatus*. Less frequent causes are *A. flavus*, *A. terreus*, and *A. niger* [9,10,11]. Aspergilli can cause mild allergies, allergic bronchopulmonary aspergillosis, chronic pulmonary aspergillosis, invasive pulmonary aspergillosis, and disseminated aspergillosis. Invasive pulmonary aspergillosis refers to a severe form of the disease that primarily affects severely immunocompromised patients, critically ill patients, and those with compromised lung function [4,5,12]. After the inhalation of conidia, an infection can either spread in the immediate area or move to other parts of the body, depending on the immune system of the host. The risk factors for this infection are constantly changing due to the emergence of many therapeutic agents that specifically target the immune system, as well as the increasing prevalence of viral infections, such as SARS-CoV-2 [13,14]. Although there have been notable improvements in the detection and treatment of aspergillosis, severe fungal illness persists and poses challenges in terms of effective therapy. The mortality rates continue to be elevated, especially among individuals with impaired immune systems. The diagnosis of aspergillosis is difficult since the current mycological test methods have limitations [15,16,17]. Additionally, the increasing numbers of reports of antifungal resistance make it even more complicated to manage the disease [18,19].

The respiratory tract is equipped with physical and anatomical barriers, such as enzymes, mucus, and epithelial cells, which promote the clearance of spores and hyphae by the adaptive as well as innate immune systems. The interaction between inhaled spores (conidia) and immune effector cells facilitates the initial immune response to *Aspergillus*-associated pulmonary illness. The stimulation of inflammatory programs, such as the NF-κB pathway and the NLRP3 inflammasome, regulates the efficient mechanisms of fungal clearance. This activation leads to a stronger production of pro-inflammatory cytokines and chemokines by epithelial cells, inflammatory monocytes, dendritic cells (DCs), and alveolar macrophages [20,21]. Aspergilli have defensive attributes (discussed in Section 2.1) that enable the fungus to try to circumvent the host’s defensive response to establish infection. Nevertheless, T helper cells’ protective capacity is mediated by the set of cytokines that regulate a defensive response during infection. However, the dysregulation of cytokines produced from T helper cells may also be implicated in invasive aspergillosis (IA) [22]. The interaction of environmental circumstances, fungal virulence factors, and the human immune response is pivotal in the development of IA. Moreover, the generation of cytokines by different immune cells is necessary to trigger the proper immune response in patients with IPA as compared with healthy individuals, as shown also in the IA mouse model [23,24]. Given that cytokines serve as crucial agents in facilitating a strong immune response and regulating both the innate and adaptive immune systems [25], the actions of cytokines have a significant impact on the host’s immunological response to *Aspergillus*. Therefore, studying the complex patterns of cytokine and chemokine responses caused by IA has revealed how IA progresses. Monitoring the fluctuations in these host molecules could potentially assist clinicians in making informed decisions about the prevention and treatment of aspergillosis, particularly in immunocompromised patients. In this review, we have summarized the role of cytokines in response to *Aspergillus* spp. infections, focusing on human, as well as animal, studies. Subsequently, the interplay of pathogen recognition patterns, chemokines, and cytokines during *Aspergillus* infection in human or animal studies is documented. We address the varied manifestations of aspergillosis with respect to this interplay.

## 2. Role of Pathogen-Associated Molecular Patterns and Pathogen Recognition Receptors during Host–*Aspergillus* Interactions

### 2.1. Pathogen-Associated Molecular Patterns (PAMPs)

PAMPs play a crucial role in the development and progression of aspergillosis. They are small molecules with a conserved motif, such as polysaccharides, present on the cell wall of microbes, but absent in the host [26]. One of the key features of PAMPs from fungal cell walls is serving as ligand molecules for the pattern recognition receptors (PRRs) present on immune cells, such as dendritic cells, macrophages, neutrophils, etc. These PAMPs primarily initiate the activation of innate immune responses, leading to the production of various pro-inflammatory cytokines and chemokines [27,28,29]. These cytokines and chemokines are essential for recruiting immune cells to the site of infection and promoting an effective immune response against *Aspergillus* infections [30]. The cell wall of *Aspergillus* spp. not only plays a crucial role in the maintenance of cell morphology, but it also helps the organism to survive unfavorable environmental stresses found in their natural niches and hosts, such as pH, temperature, osmotic pressure, nutrient level, and hypoxia. Aspergilli respond to these challenges in various ways, which include rebuilding and repairing their cell walls [20,31]. The cell wall component of aspergilli interacts with host immune defenses [32] and host immune cells. Therefore, cell wall remodeling is a crucial mechanism used by aspergilli in response to the host environment and defense system. The cell wall of *Aspergillus* is mainly composed of polysaccharides [31]. Moreover, the transition of *Aspergillus* spp. conidia to germinating conidia, hyphae, and mycelium leads to changes in the cell wall structure, and these morphotypes interact with the host environment and host defense system [33].

*Aspergillus* species have distinct PAMPs, which are usually present on the cell wall [31]. These components in their conidia and hyphae are glucans, chitins, and galactomannan. *Aspergillus* spp. are characterized by different morphotypes, a dormant stage or quiescent morphotype, followed by the germinative type and hyphal type or mycelium type, collectively called vegetative morphotypes [34]. Over 90% of the hyphal cell wall component of *Aspergillus* spp. (e.g., *A. fumigatus*) consists of complex carbohydrates that are covalently bonded. The main polysaccharides include α-glucans (mainly α-1,3-glucan, which also includes small quantities of α-1,4-glucan), β-glucans (β-1,3-glucan with β-1,6-branches), galactomannan, and chitin in the cell wall [35]. β-1-3-glucan is linked to chitin and galactomannan, whereas α-1-3-glucan and galactosaminogalactan (GAG) fill the spaces between fibrillar polysaccharides. Interspersed galactomannan is a polysaccharide that consists of a mannose backbone with side chains of galactose. Another component, chitin, is a long-chain polymer of N-acetylglucosamine, a derivative of glucose [36,37].

The components of conidia and hyphal forms that act as PAMPs differ in their outer layers. With the onset of germination, a conidium loses its hydrophobic rodlet and melanin layer, and the previously hidden cell wall polysaccharides are revealed. The rodlet layer, formed of hydrophobins, contributes to immune evasion by encouraging the adherence of conidia and biofilm formation, allowing the persistence of conidia within the host [38,39]. The melanin of conidia is involved in scavenging reactive oxygen species (ROS) and limiting the generation of nitric oxide by phagocytic cells [40]. Thus, the rodlet and melanin layers of conidia help in escaping or weakening the innate effectors’ responses. The polysaccharide component, α-1-3-glucan, moves from the inner layer to the outer layer of the surface. Thereafter, galactosaminogalactan appears on the cell wall’s surface, which is involved in intrahyphal adhesion [36,38]. The hyphal cell wall composition changes according to the environmental conditions (hypoxia, microbiota, pH, drugs, etc.) prevailing in different tissues or organs during infection. Under the conditions that prevail in the lungs, the *A. fumigatus* cell wall has less β-(1,3)-glucan, whereas under hypoxic conditions, there is an increase in β-(1,3)-glucan and chitin and a reduction in α-(1,3)-glucans [41]. Thus, constant interaction with the host immune response is essential to recognize *Aspergillus* species PAMPs, as their cell wall continuously remodels during the initial stages of infection and invasion of host tissues. The prominent PAMPs associated with the cell wall of aspergilli in the conidial to hyphal forms are illustrated in Figure 1A,B.

### 2.2. Pathogen Recognition Receptors (PRRs)

PRRs are a crucial component of the innate immune system, playing a key role in the recognition of pathogens and initiation of the immune response. PRRs are categorized into soluble and cell surface-bound receptors. Soluble receptors, such as collectins (which include MBL, SP, and CL-11), as well as PTX-3 and ficolins, are reported to be important for the recognition of *Aspergillus* spp. Apart from these, cell-surface-bound receptors, such as Toll-like receptors (TLRs) and dectins, also play significant roles in the recognition of aspergilli and initiate the immune response against them [42]. PRRs on various cells, such as macrophages, dendritic cells, and epithelial cells, recognize PAMPs for the initiation of an effective immune response against fungal infections. Upon activation, PRRs lead to the recruitment and activation of immune cells, phagocytosis of fungal pathogens, and the production of antimicrobial peptides and cytokines. Understanding the interplay between PRRs and fungal pathogens is essential for developing strategies to modulate the immune response and improve antifungal therapies. There are several families of PRRs, and these include TLRs and C-type lectin receptors (CLRs). TLRs are transmembrane proteins that are primarily expressed on the cell surface or within endosomes, including TLR-1, 2, 3, 4, 6, 9, and 13. TLRs, especially TLR2 and TLR4, are involved in recognizing fungal cell wall components. TLR2 can recognize lipoproteins, while TLR4 can recognize the glycans of aspergilli. The activation of TLRs initiates signaling cascades that contribute to the immune response against *Aspergillus* infections. CLRs are involved in the recognition of fungal and bacterial pathogens. They are expressed on the surface of immune cells and can bind to carbohydrates present on the surface of microbes. One of the primary CLRs involved in the recognition of *Aspergillus* spp. is Dectin-1 (Dendritic Cell-associated C-type Lectin-1). Dectin-1 is a transmembrane receptor expressed on the surface of immune cells, such as macrophages, dendritic cells, and neutrophils. It specifically recognizes the β-glucans of *Aspergillus*. Dectin-2 is another member of the CLR family and is involved in the recognition of mannose-rich molecules and galactomannan on the *Aspergillus* cell wall [43]. Host mannose-binding lectin (MBL) is also involved in interactions with *Aspergillus* during infection. These interactions may include the activation of MBL-associated serine proteases, MBL-mediated activation of complement, opsonization, and an effect on inflammatory mediators. The activation of complement results in components increasing the influx of PMNs to the site of infection, and the components may participate in antifungal activity. Resting conidia trigger the alternative complement pathway. As conidia start to swell and are transformed into hyphae, the classical pathway dominates [44]. A study showed that the MBL of the lectin complement pathway binds to carbohydrate moieties on the *Aspergillus* cell wall and activates complement pathways by increasing the deposition of the C4 component of the complement pathway. Furthermore, MBL can support C3 cleavage by a C2 bypass mechanism after contact with *A. fumigatus* conidia, resulting in the activation of the alternative pathway and avoiding the formation of the classical pathway C3 convertase [45]. Kaur et al. [46] demonstrated that recombinant MBL treatment of mice with pulmonary aspergillosis increased the levels of tumor necrosis factor (TNF)-α and IL-1α and significantly decreased IL-10, enhancing survival and reducing the fungal burden. In studies with systemic murine aspergillosis, our group found that MBL-knockout mice were more resistant to disease than wild-type MBL-sufficient mice [47], which is suggestive of a deleterious MBL effect owing to increased PMN influx and the uncontrolled discharge of reactive oxygen species into the tissues.

β-glucan, a glucose polymer that forms a major part of the fungal cell wall of aspergilli [48], has been found to specifically suppress the TLR-4 induced response in the host immune system, whereas α-glucan has been found to inhibit IL-6 production induced through TLR-2 and TLR-4 stimulation [29]. Galactomannan has been found to diminish TLR-4 mediated responses, while its inhibitory effects on TLR-2 signaling are limited [49,50]. According to studies, chitin has pluripotent immunomodulatory capabilities [49,51,52,53], and the role of downstream signaling pathways needs to be delineated to ascertain how they might affect lung pathologic pictures. The full panoply of chitin receptors may not yet be understood. We discuss the role of chitin further below. An effort to better understand the role of chitin could prove important, since chitin synthesis inhibitors are available (and also act synergistically with β-glucan inhibitors), and could be used in therapy in the future [54].

When conidial germination occurs, Dectin-1 identifies β-1,3-glucan, which is particularly important in germinating conidia and immature hyphae, owing to the increased exposure of β-1,3-glucans, compared with mature hyphae, where β-1,3-glucans are concealed by exopolysaccharides [55,56]. This recognition occurs only when β-1,3-glucan is in fibrillary or particulate forms, and not when it is in a soluble form [57,58]. Dectin-1-dependent responses are particularly significant. The recognition of various aspergilli PAMPs with associated PRRs is illustrated in Figure 2.

GAG is an important exopolysaccharide that acts as an adhesin, enabling the attachment of hyphae to macrophages, neutrophils, natural killer (NK) cells, and platelets [57]. CD56, another PRR [65], is primarily expressed on NK cells and interacts with the cell wall component, GAG, of *A. fumigatus*. This interaction leads to the activation of NK cells, degranulation, and the production of effector molecules, such as IFN-γ and TNF-α [66]. Another human cell receptor is DC-SIGN, which acts as an adhesion receptor that exclusively binds to the galactomannans found in the cell wall of *A. fumigatus* [67]. Dectin-2 detects α-mannans, thus playing a vital role in the binding of swollen conidia and hyphae to the immune cells, particularly macrophages [43]. E-cadherin on type II pneumocytes can mediate the internalization of *A. fumigatus* conidia [68]. Another PRR, Ephrin type-A receptor 2 (EphA2), is present on oral and respiratory epithelial cells, and it binds to β-glucans during the remodeling of the conidial cell wall [69]. EphA2 signaling has been implicated in phagocytosis by alveolar macrophages and epithelial cells, thereby promoting the clearance of conidia, and the presence of DHN melanin on the conidia induces EphA2-dependent internalization of *A. fumigatus* conidia [70]. In addition, EphA2 activation stimulates the production of pro-inflammatory cytokines and chemokines, contributing to the recruitment and activation of other immune cells to the site of infection [70]. Chitin is a crucial structural element found in the cell wall of fungi and is the second-most-prevalent polysaccharide in nature. Several studies have discovered that administering chitin particles through the nose is enough to induce type-2 inflammation [71,72]. This indicates that chitin may play a crucial role in triggering allergic reactions. Furthermore, chitin extracts from house dust, containing *Aspergillus*, were able to induce type 2 immune responses in mice [73]. Nevertheless, there is a paucity of experimental evidence supporting the notion of how chitin triggers type-2 immunity. However, the co-recognition of chitin along with β-glucan has been demonstrated in some studies [61,74]. Becker et al. [49] observed that chitin induced an anti-inflammatory signature characterized by the production of IL-1Ra in the presence of human serum, the production of which was abrogated in immunoglobulin-depleted serum. The Fcγ-receptor-dependent recognition and phagocytosis of IgG-opsonized chitin was identified as a novel IL-1Ra-inducing mechanism [49]. Wagener et al. [61] showed that digested chitin particles are phagocytosed via the mannose receptor, and NOD2 and TLR-9 co-localize with internalized chitin, leading to the production of IL-10. Moreover, He et al. demonstrated that LYSMD3, a receptor expressed on the surface of human airway epithelial cells, can bind chitin, as well as β-glucan, on the cell walls of fungi [21,74], and subsequently, *A. fumigatus* spores are taken up by airway epithelial cells [75]. The precise mechanism by which chitin initiates allergic inflammation remains unclear, and the outcomes vary depending on the dimensions of the chitin particles employed [76]. More studies are required to investigate the role of chitin and its recognition during *Aspergillus* infection, especially ABPA. 

Apart from surface-bound and endosomal PRRs, cytosolic PRRs, such as NOD-like receptor (NLR), are also stimulated by PAMPs [77]. The NLR family includes nucleotide-binding oligomerization domain 1 (NOD1), NOD2, NLRP3/Cryopyrin/Nalp3 ‘inflammasome’, and NLRC4/Ipaf inflammasome [78,79]. A study on human monocyte cell lines discovered that *Aspergillus* hyphal fragments induce NLRP3 inflammasome assembly and IL-1β cytokine production. Further, a study showed that *A. fumigatus* conidia increased NOD2 expression in alveolar epithelial cells and macrophages as well as in the lungs of mice in a pulmonary invasive aspergillosis model [80].

Moreover, retinoic acid-inducible gene-1 encodes (RiG-1)-like receptor (RLRs), another intracellular PRR that recognizes PAMPs. RLRs include RIG-1, melanoma differentiation-associated 5 (MDA5), and laboratory genetics and physiology 2 (LGP2). A study on *A. fumigatus* demonstrated that MDA5/MAVS (mitochondrial antiviral signaling) signaling is essential to resist pulmonary *A. fumigatus* infection in a murine model. MDA5/MAVS activates in response to the ds-RNA of live *A. fumigatus* and induces type III interferon (IFN) expression and the production of the CXCL10 chemokine. It has also been demonstrated that the neutrophil killing of *Aspergillus* conidia depends upon MDA5/MAVS signaling [81].

Another study demonstrated that genetic polymorphisms in MAVS alter the production of chemokines, creating a risk to the patient for invasive pulmonary aspergillosis. In a mouse model of *A. fumigatus* infection, it was also demonstrated that alveolar macrophages are the key cells where the MDA5/MAVS-dependent interferon response is induced [82].

## 3. Role of Cytokines, Chemokines, and Immune Cells during *Aspergillosis*

The immune response to invasive aspergillosis (IA) involves a complex interplay of innate and adaptive immune responses. *Aspergillus* proteases disturb airway epithelium and initiate host secretory responses [83,84]. The mammalian host mobilizes molecules (e.g., transferrin) to deny iron from the invader and neutralize fungal attempts to acquire iron via fungal siderophores for microbial sustenance [85]. Innate host factors such as mannose-binding lectins, defensins, surfactant proteins, and pentraxin also represent a first line of host defenses against invading pathogens [64,85,86,87,88,89]. Host immunogenetic factors can prove to be determinants of the outcome in these first steps, and the ones to follow [90]. In subsequent Section 3.1 and Section 3.2, we review the sequential events that occur in invasive aspergillosis or allergic manifestations of aspergillosis [91].

Cytokines play a central role in the immune response, serving as signaling molecules that facilitate communication among immune cells and coordinate the various aspects of the immune system. These small proteins are produced by a wide range of cells, including macrophages, T cells, and B cells. The immune response involves a complex network of interactions, and cytokines act as messengers that regulate the activities of immune cells. These activated immune cells can then eliminate pathogens. In the case of aspergillosis, these molecules play a crucial role in protecting the host from infection by facilitating the start, direction, continuation, and progression of the host’s response [24]. The initial response to *Aspergillus* infection is carried out by innate immune cells, which include dendritic cells (DCs), macrophages, and airway cells, including alveolar epithelial cells (AECs). Moreover, macrophages engulf *Aspergillus* conidia through phagocytosis and prevent their germination during the initial stage of infection [92]. This process triggers the production of inflammatory chemokines and cytokines. In addition, neutrophils and circulating monocytes cause harm to hyphae by releasing both oxidative and non-oxidative microbicidal chemicals [93,94,95,96]. Although *Aspergillus* is encountered by the majority of humans without any negative consequences, certain individuals, particularly those with compromised immune systems or pre-existing health disorders, may be vulnerable to *Aspergillus* infections. A study on lung infection with *A. fumigatus* revealed that, during infection, inflammatory monocytes can transform into dendritic cells [97]. These cells then displace the *A. fumigatus* conidia to the nearby lymph nodes that are responsible for draining the lungs. This connection between the monocyte-derived dendritic cells and the lymph nodes plays a crucial role in the development of acquired immunity and activation of T cells [97].

### 3.1. The Th2 Response and Allergic Bronchopulmonary Aspergillosis (ABPA) and Chronic Aspergillosis

ABPA is a hypersensitive response to the fungus *Aspergillus*, predominantly impacting the respiratory system. It is distinguished by an excessive immunological reaction to the presence of *Aspergillus* spp. in the respiratory tract, resulting in inflammation and respiratory symptoms. Gaining insight into the precise role of various cytokines and chemokines in ABPA is crucial for the advancement of focused treatments. Chronic pulmonary aspergillosis (CPA) is a gradually advancing aspergillosis that primarily impacts patients with pre-existing lung diseases, such as chronic obstructive pulmonary disease (COPD) or lung cavities caused by past infections.

ABPA is linked to an immunological response that is mediated by Th2 cells. T-helper cells, particularly Th2 cells, secrete cytokines, such as interleukin (IL)-4, IL-5, and IL-13. These cytokines play a role in stimulating eosinophils and mast cells, which leads to the development of the allergic and inflammatory responses observed in ABPA. The Th2 response, which involves the secretion of IL-4, IL-5, IL-13, and IL-10, is responsible for promoting anti-inflammatory reactions, allergic responses, and the persistence of fungal infections in the lungs [98]. The importance of a shift towards Th1, rather than Th2, for providing protection is particularly emphasized in mice lacking IFN-γ: these mice exhibit compromised antifungal immunity and increased Th2 responses [99]. An intense Th2 immune response has been linked to a negative outcome of aspergillosis in murine models [100], mediated by the cytokine IL-4, and this response counteracts the beneficial Th1 immune responses [99]. Furthermore, the release of IL-10 by Th2 cells has a detrimental effect on protective Th1 responses in aspergillosis models. This detrimental effect is achieved by the suppression of pro-inflammatory cytokines and chemokines, the inhibition of T-cell activation and IFN-γ production, and the promotion of a Th2 response. A gene expression study on human monocytes co-cultured with *A. fumigatus* conidia observed elevated expression of genes coding for IL-1*β*, IL-8, CXCL2, CCL4, CCL3, and CCL20. This upregulation coincided with an observed increase in phagocytosis [101]. In a separate investigation employing identical co-incubation methodology, it was observed that 602 genes were differentially regulated. Monocytes exhibited increased production of IL-8, CCL20, and CCL2 after 3 h of co-incubation with *A. fumigatus* hyphae, but only 206 genes were affected in response to resting conidia [102]. In neutropenic mice, it has been observed that natural killer cell adoptive transfer led to a decrease in the expression of CCL2 in the lungs at the initial stage of infection. Additionally, when this protein was neutralized, there was a higher death rate and a greater amount of fungal presence in the lungs [103].

The stimulation of bronchial epithelial cells with *A. fumigatus* triggers the activation of protease receptor (PAR-2) and PTPN11 (SHP2), which is a phosphatase that hinders IFN-γ signaling. As a result, this causes a preference for a Th2 cell response. IL-33, a cytokine belonging to the IL-1 family and recognized for activating the Th2 immune response, is abundantly produced in mice exposed to viable *A. fumigatus* conidia. IL-33 is well recognized for its role in causing immune-related tissue damage in the respiratory system due to prolonged exposure to allergens [104]. The analysis of gene expression showed that several genes associated with cytokines and chemokines, such as CCR4, CCR5, and CCL27, were more active in a mouse model of acute ABPA compared with control animals [105]. In addition, the levels of CCL2, CCL3, CCL5, CCL6, CCL11, CCL17, CCL22, and CXCL1 in the lungs were elevated during experimental fungal asthma [106,107]. Compelling evidence supports the role of chemokines, specifically CCR4 and its ligands, in the development of ABPA [108]. Furthermore, it has been demonstrated that CCL17 and CCL22 exhibit significant increases following exposure to conidia in CCR4-positive mice that have been sensitized to *A. fumigatus*. In comparison, in CCR4-negative mice, a strong antifungal immune response was characterized by enhanced neutrophil and macrophage recruitment [106]. Therefore, it is suggestive that CCL17 and CCL22 have important roles in ABPA and can be targeted for future therapy. In human cystic fibrosis ABPA patients, the CCL17 level was reported to be elevated and characterized by ABPA exacerbations [108]. CCL17 may aid in dampening the Th2 response and allow the triggering of Th1 and Th17 responses against ABPA. In a study at Nagasaki University Hospital between January 2003 and December 2018, 14 individuals without any health issues, 19 individuals diagnosed with asthma, 11 individuals with allergic bronchopulmonary aspergillosis (ABPA), and 10 individuals with chronic pulmonary aspergillosis were included to understand the serum cytokine profiles in the pathology of ABPA and CPA. Patients with ABPA exhibited markedly elevated levels of IL-5, whereas patients with CPA showed significantly elevated levels of IL-33 and tumor necrosis factor (TNF) [109]. Further study revealed that *Aspergillus* conidia elicit Th2 responses in human peripheral blood mononuclear cells (PBMCs) through a route that depends on the complement receptor, CR3. Also, increased levels of IL-5 and IL-13 were observed, as well as a decrease in IFN-γ [110]. IFN-γ can be utilized to modulate the T helper response to Th1 type during ABPA.

In ABPA, an increased presence of lung neutrophils was observed [111]. Additionally, ABPA patients exhibited high levels of matrix metalloproteinase-9 (MMP-9) [112]. Elevated IL-8 levels correlated with the level of neutrophils and MMP-9 in ABPA patients’ sputum compared with controls [112]. This suggests that the recruitment of neutrophils is regulated by IL-8, and the result is tissue damage in the lungs through the action of the MMP-9 enzyme.

The management of ABPA often entails a synergistic administration of antifungal agents (to regulate the proliferation of *Aspergillus*), corticosteroids (to mitigate inflammation), and occasionally immunomodulatory therapies. Vigilant observation and continuous control are crucial, as ABPA might exhibit an unpredictable clinical progression, with the possibility of exacerbations. Gaining a comprehensive understanding of the precise cytokine and chemokine reactions in ABPA is crucial for the development of focused therapeutic interventions. For example, in the treatment of ABPA, it may be beneficial to use drugs that inhibit certain cytokines that are involved in the Th2 immune response, such as the antagonism of IL-5, IL-13, and chemokine CCR4, as well as modulation of the Th2 response towards Th1/Th17.

The Th9 subset has a close relationship with the Th2 immune response [113]. Th9 cells are responsible for promoting inflammation, infection, and allergies [114]. This specific type of immune cell is formed when exposed to the Th2-polarizing cytokine IL-4, along with IL-2 and TGF-β. When fully specialized Th2 cells are cultivated with TGF-β, there is an enhanced synthesis of IL-9, which is a distinctive cytokine of Th9 cells [115]. The Th9 subset of T cells has a significant impact on the allergic response to *A. fumigatus* in individuals with cystic fibrosis [116]. In this context, inhibiting the Th9 response could serve as an innovative treatment approach to decrease inflammation associated with infection. Th9 inhibitors have been identified, and may lead to therapeutics against allergic lung diseases, as has been the case with Th2 monoclonal antibody inhibitors.

### 3.2. Invasive Aspergillosis

Invasive aspergillosis (IA) primarily affects individuals with weakened immune systems, such as those undergoing chemotherapy, organ transplant recipients, or individuals with advanced HIV/AIDS [117]. The immune response to invasive aspergillosis involves a complex interplay of various cytokines and immune cells. During the initial phases of invasive aspergillosis, immune cells generate proinflammatory cytokines, including TNF-α, IL-1, and IL-6. These cytokines have a function in stimulating the immune response and attracting additional immune cells to the location of the infection [118]. The adaptive immune response is initiated through the activation of T-helper cells, specifically Th1 and Th17 cells. Th1 cells secrete cytokines, such as interferon-gamma (IFN-γ), which play a crucial role in stimulating macrophages and boost their ability to combat *Aspergillus* infections [119,120]. Th17 cells secrete IL-17, which aids in the mobilization of neutrophils, a vital leukocyte involved in the immune response to invasive *Aspergillus* infection [121,122,123].

In invasive pulmonary aspergillosis, the lung’s resident alveolar macrophages (AMs) and epithelial cells engage in initial interactions with developing *Aspergillus* conidia [117]. These cells detect PAMPs present on the surface of aspergilli, facilitated by PRRs such as TLR-1, 2, 3, 4, 6 [124,125,126,127]. Also, C-type lectin receptors Dectin-1 and Dectin-2, or NOD-like receptors, are involved in recognition. The identification of *Aspergillus* triggers the production of pro-inflammatory cytokines, such as IL-1α, IL-1β, TNF-α, IL-8, and MIP-1α, through the activation of the NF-κB and inflammasome pathways [27,125,128,129,130].

Alveolar macrophages engulf aspergilli conidia and eliminate them by acidifying the phagolysosome and activating antimicrobial enzymes (cathepsin D and chitinase), as well as generating reactive oxygen species (ROS) [93,131]. Chemokines and proinflammatory cytokines function as chemoattractants and stimulators for various immune cells at the site of infection. They recruit neutrophils, natural killer cells, and T lymphocytes, and thus prevent the colonization of the host. Following the activation of lung epithelial cells and resident alveolar macrophages, neutrophils are among the initial cells to migrate to the site of infection to commence immunological responses against *Aspergillus* spp. in both mouse models and humans [132,133,134]. The predominant factor contributing to this role was observed in patients who received medicines that induced neutropenia, as well as in those who had mutations in molecules involved in neutrophil activity, such as NADPH oxidases [133,134]. Neutrophils are drawn to the location of infection by IL-8 and IL-17. However, the precise function of IL-17 in the context of IA is unclear [22,60,135,136], which we will discuss in more detail subsequently. In addition to amplifying the inflammatory response through the secretion of cytokines and chemokines, neutrophils can directly engulf and destroy the fungus by generating ROS and antimicrobial chemicals [137]. Neutrophils possess an additional antifungal mechanism known as neutrophil extracellular traps (NETs), although it is not entirely clear that NETs are always beneficial. Using an animal model of aspergillosis, the activation of NADPH oxidase in neutrophils sheds light on the role of NETs in the containment and moderate clearance of *Aspergillus* hyphae [138,139].

Dectin-1, as mentioned above (Section 2.2), is a C-type lectin protein that is one of the primary innate receptors present on immune cells that recognize *A. fumigatus,* especially when β-glucans are exposed during cell wall remodeling, and is protective [59]. It has been observed that dectin-1 deficient mice exhibit increased vulnerability to fungal infection compared with the control group. These mice have demonstrated impaired production of inflammatory cytokines and chemokines, including IL-1β, TNF-α, CCL3, CCL4, and CXCL1 [60]. Consequently, there is insufficient recruitment of neutrophils to the lungs and inadequate production of ROS, which contributes to the uncontrolled growth of *A. fumigatus* in the lungs.

Reedy et al. demonstrated that the dectin-2 receptor recognizes the galactomannan of *Aspergillus* and induces the production of the proinflammatory cytokine TNF-α in mouse lungs [43]. It has been observed that dectin-2-deficient mice, although having increased immune cell ingress to the lungs, are not protected [43]. A study with human-derived peripheral blood mononuclear cells having impaired dectin-2 receptors observed the diminished production of TNF-α and IL-6 in response to *A. fumigatus*. This factor in invasive aspergillosis apparently led to the death of a patient [140]. Therefore, dectin-2 could be an important vector to trigger proinflammatory cytokine responses and clearance of aspergilli, and avoid invasive aspergillosis.

Studies on TLRs by Rubino et al. [141] demonstrated that TLR2 and TLR4 played a crucial role in the recognition of *A. fumigatus* by innate immune cells in both human and murine cells. Using a rodA knockout strain of *A. fumigatus*, they observed the failure of *A. fumigatus* recognition by innate immune cells by TLR2 and TLR4. Moreover, the recognition was independent of TLR1 in human and mouse cells and of TLR6 in murine cells [141]. The bone-marrow-derived macrophages from mice lacking TLR1, TLR2, TLR3, TLR4, and TLR6 were cultured together with the fungus. TLR1-deficient macrophages revealed a decline in the production of proinflammatory cytokines and chemokines, including IL-12p40, CXCL2, IL-6, and TNF-*α*. This reduction in production was nearly eliminated in TLR2, TLR4, and TLR6 knockout cells, but not in wild-type or TLR3-deficient macrophages. Furthermore, the activation of TLR9 was found to be associated with the presence of unmethylated CpG patterns on the DNA of the fungus [62]. Hence, the significance of TLR is evident, particularly TLR1, TLR2, TLR3, TLR4, TLR6, and TLR9 in the identification and response to *A. fumigatus* by cells of the innate immune system [62,141,142].

More investigations in mice revealed that *A. fumigatus* may trigger any of the Th1, Th2, Th9, Th17, or Th22 CD4 T-cell responses, based on the specific conditions of antigen exposure. In experiments, the occurrence of invasive *A. fumigatus* infection was increased when the activity of IFN-γ or TNF-α was suppressed, indicating that Th1 cells play a protective role against fungal invasion [143,144].

Immunotherapy, such as that with IFN-γ, shows potential, particularly in patients with impaired immune systems [145]. This occurs through the activation of macrophages that boosts their antifungal activity [146]. IFN-γ promotes Th1 cell development and increases the production of antifungal cytokines, including IL-12 and TNF-α [146,147]. The exogenous treatment of IFN-γ can enhance the host’s immune response against *Aspergillus*, leading to better outcomes in some individuals in some clinical research studies [148]. IFN-γ may synergize with conventional antifungal therapy [149].

TNF-α, like IFN-γ, activates PMNs, leading to increased oxygen radical release and hyphal injury to *A. fumigatus* in in vitro studies [143,150]. Apart from these cytokines, colony-stimulating factors (G-CSF, GM-CSF and M-CSF) have been reported to enhance the capacity of the innate immune response against mycoses, including invasive aspergillosis [151,152,153].

Recruitment cytokines are essential for facilitating the migration of particular leukocytes to the location of infection in invasive aspergillosis. The cytokines that are crucial for protective responses against infection are linked to Th-1 cells, such as IL-12, IL-18, and IFN-γ. In contrast, the cytokines IL-4 and IL-10 of the Th2 cells play a role in the advancement of the infection. A study in immunocompromised mice observed a significant increase in the TNF-α levels in the lungs when exposed to *Aspergillus* conidia. Within this group of recruitment cytokines, the subset of CXC chemokines and their receptor CXCR2 plays a crucial role. These chemokines aid in attracting neutrophils, whereas chemokines CCL3 and CCL2 are essential for attracting monocyte-lineage leukocytes and NK cells. Neutralizing TNF-α in these mice leads to a decrease in their ability to eliminate the fungal infection and an increase in mortality rates [93]. The inhalation of *A. fumigatus* conidia in mice leads to the production of IL-12, IL-18, and IFN-γ in the bronchoalveolar lavage fluid and the lung tissue [154]. In mice, when these cytokines are removed using neutralizing antibodies, the clearance of *A. fumigatus* from the lungs is delayed [154]. It has been observed that the depletion of TNF-α can lead to reduced activity of myeloperoxidase in the lungs and a decrease in the synthesis of chemokines that attract inflammatory cells to the lung. In contrast, administering a TNF-α agonist directly into the trachea of immunocompromised animals before infection significantly reduced the severity of the infection [155]. Similarly, a reduction in the severity of infection was also observed in immunocompetent animals, but only with large *Aspergillus* inocula [155], indicating that TNF-α is a critical component of innate immunity in both immunocompromised and immunocompetent hosts [143]. Additionally, it accounts for the sporadic development of invasive aspergillosis in patients undergoing treatment with TNF-α antagonist medicines.

There is a positive correlation between the production of Th1 cytokines and improved outcomes of IA, whereas the production of Th2 mediators (a pathway described in detail in Section 3.1) is associated with more severe illness. The kinetics of pro-inflammatory cytokines in the inhaled model of IA in mice showed a sequential increase in IL-18, IL-12, and IFN-γ in the bronchoalveolar lavage (BAL) fluid and lung homogenates [144,154,156]. In contrast to the stimulation of IL-12 and IFN-γ mRNAs, the stimulation of IL-18 protein occurred without the stimulation of IL-18 mRNA, an indirect indication of a post-transcriptional mechanism involved in the processing of IL-18. The administration of neutralizing antibodies to cytokines IL-18 and IL-12 exacerbated infection in mice. These findings suggested that the cytokines IL-18 and IL-12 within the lungs are crucial for eliminating the infection [154].

Another study revealed that mice lacking IL-4 are more resistant to IA, as evidenced by decreased lung inflammation and decreased fungal growth [106]. In addition, IL-4 deficient mice showed an increase in Th-1 type cytokines characterized as IL-12 and IFN-γ. Furthermore, their vulnerability to IA infection was enhanced when they were treated with an IL-12-neutralizing antibody [99]. These data suggest that IL-4 suppresses the Th1 cell response and plays a role in the development of IA. In contrast, regarding IL-10, an interleukin mentioned above (Section 3.1), a study showed that mice lacking this cytokine exhibited an antifungal inflammatory response; there was an increased production of Th1-type cytokines and a decreased severity of infection [94]. Mice lacking IL-10 showed resistance to infection when conidia were administered intravenously [95]. Mice that do not manufacture the IL-10 cytokine are more resistant to fungal invasion [157,158]. When IL-10 is removed during the challenge, it improves the elimination of the fungus by boosting the production of IL-12 and IFN-γ, cytokines that have a beneficial effect in fighting against *A. fumigatus* [99]. IL-10 reduces the ability of macrophages to destroy *A. fumigatus* conidia by hindering the oxidative burst at the macrophage level [157].

Studies have shown that mice lacking IL-6 are more prone to invasive pulmonary aspergillosis. The findings revealed an increase in fungal growth and a lower survival rate among these animals [159]. The BAL fluids of IL-6 knockout mice showed elevated concentrations of IL-17 and IL-13 cytokines. Research has demonstrated that IL-17 stimulates the release of cytokines responsible for the movement of neutrophils from human airway cells [160]. The increased expression of this cytokine may serve as an additional mechanism for attracting neutrophils to the lungs in this *Aspergillus* infection scenario. A study on repeated exposure to *A. fumigatus* conidia in mice revealed 99% conidial clearance after 24 h [16,161]. Also, after two challenges, there was an influx of neutrophil and T-regulatory cells in lungs with less inflammation. This study showed the co-evolution of Th1, Th2, and Th17 responses in lungs with repeated exposure to *A. fumigatus*. Cytokines IL-1, IL-6, and IL-23 initiate the differentiation of Th17. IL-1, IL-36α, β, γ, and IL-36Ra also regulate the Th17 response in the case of *Aspergillus* infection. IL-17 is the hallmark of the production of Th17 cells and also triggers the activation of neutrophils to the site of infection. IL-17 induces the proinflammatory cytokines IL-6, IL-1β, G-CSF, and TNFα, and chemokines CXCL8, MIP-1, and MCP1 [161]. However, it has been observed that the increased Th17 response can lead to severe immunopathology, characterized by more neutrophil infiltration into the lungs and inadequate fungal clearance. In addition, IL-23 has been reported to be important for the induction of Th17. However, it is not essential for initiating Th17 differentiation, but required for the maintenance of Th17 differentiation [162]. Moreover, in the mouse model of IA, the absence of IL-12 led to enhanced production of IL-23 and high susceptibility to pulmonary aspergillosis. Neutralizing antibodies against IL-23 and IL-17 lead to a decrease in *Aspergillus* burden in the IA murine model [16,163]. IL-22, another member of the Th17 group, belongs to the IL-10 family and has demonstrated significant involvement in various illnesses, such as bacterial pneumonia and lung infection caused by *Aspergillus* spp. In *A. flavus* keratitis, elevated levels of IL-22 expression were detected in both patients and an experimental mouse model of corneal infections [164]. Concurrently, IL-17, IL-23, and IL-18 production was also observed. IL-22 plays a crucial role in promoting inflammation in the presence of microbial infections, especially targeting fibroblasts and epithelial cells. IL-22 receptors were consistently expressed in human corneal endothelial cells (HCECs), and IL-22 triggered the activation of NF-κB and MAPK pathways, leading to antimicrobial peptide production. In addition, the use of recombinant IL-22 decreased the amount of fungus and the cloudiness of the cornea in the mouse model of *A. flavus* keratitis [164]. These observations suggested a multifaced role of Th17 cells in response to *Aspergillus* spp. infections [60,157,159,161]. Using RNA-seq, Shankar et al. observed the upregulation of genes encoding dectin-1, TLR-2, TLR-3, TLR-8, TLR-9, TLR-13, and soluble receptors, such as Ptx-3 and C-reactive protein genes; concurrently, they revealed Th1 and Th17-type immunity in the kidneys of a mouse model of aspergillosis [165].

IL-27 is a significant regulatory cytokine, mostly recognized for its diverse effects on T cells [166,167]. Initially, IL27 was characterized as pro-inflammatory owing to its ability to increase the expression of T-bet in CD41 T cells and facilitate the development of Th1 cells [168,169]. Subsequently, IL-27 was found to have anti-inflammatory properties and IL-27 could inhibit Th1, Th2, and Th17 responses, as well as the formation of dendritic cells [166,170,171]. A recent study revealed that this cytokine had a pro-inflammatory effect on mice that were repeatedly infected with *A. fumigatus* [16]. When mice were exposed to *A. fumigatus,* there was a notable increase in the production of IL-27 in their lungs. The elimination of the *IL-27Ra* gene in mice led to a prominent increase in the number of fungi in the lungs during infection. The heightened fungal proliferation in *IL-27Ra*-deficient mice was linked to the decreased production of IL-12, TNF-α, and IFN-γ, decreased expression of T-bet, as well as a decrease in CD4^+^ T cells and their activation in the lung. This indicates that IL-27 signaling fosters Th1 immune responses after repeated exposure to *A. fumigatus*. Furthermore, *IL-27Ra*-deficient infected mice showed a decrease in the accumulation of dendritic cells and exudate macrophages in their lungs. Also, these cells exhibited a reduced expression of MHC class II. Hence, it can be concluded that IL-27 plays a crucial role in promoting type-1 immunity and is essential for suppressing the growth of fungi in the lungs of mice that are repeatedly exposed to *A. fumigatus*. This further emphasizes the protective function of IL-27 in the context of fungal infections [172].

Recent studies focusing on fungal extracellular vesicles (EVs) have shown their role as virulence factors and they play a vital part in the infection process. A study on *A. flavus* extracellular vesicles (EVs) found that they stimulated macrophages to generate inflammatory mediators, including nitric oxide, TNF-α, interleukin-6 (IL-6), and IL-1 [16,173]. In addition, the *A*. *flavus* extracellular vesicles improved the process of phagocytosis and killing by macrophages, and they also stimulated the polarization of M1 macrophages under in vitro conditions [173]. *A. flavus* brain infection in mice showed elevated secretion of IFN-γ, IL-12p40, and IL-6, while simultaneously showing decreases in the secretion of the Th2 cytokine IL-4 and the Th17-boosting cytokine IL-23 during the later stage of infection. Mice exposed to *A. flavus* developed inflammatory granulomatous cerebral aspergillosis, which was characterized by a significant elevation in Th1 cytokines and the occurrence of necrosis of neurons [174]. There was no significant alteration observed in the levels of pro-inflammatory cytokines, such as TNF-α, in the brain tissue at various time intervals during our investigation. Nevertheless, the proliferation of hyphae likely prompted the microglia to become activated, resulting in the local release of TNF-α. Activated microglial cells and monocytes secrete IL-12 that simultaneously induces the release of IFN-γ by T-cells. This leads to the establishment of a Th1 type of cellular immune response to the ongoing infection [174].

## 4. Conclusions and Future Perspectives

Fungi are eukaryotic organisms with complex life cycles. They synthesize or produce a wide range of molecules during their life span from conidia to germinating conidia and hyphae to mycelium. From the conidia to the hyphal phase, they remodel their cell wall and adapt to the surrounding environment. Like the complex nature of fungi, the human immune response to fungi is complex. This nature is accompanied by various components of the immune system involving both innate and adaptive immunity. The protective immune response demonstrated by Th1 and Th17-type immunity to aspergilli and the destructive response by Th2-type immunity have been noted. Presently the literature suggests many gaps in current knowledge regarding the immunity type in the case of *Aspergillus* infections. For example, the question is still open for defining Th17-type immunity; some studies have suggested a multifaceted role of both anti-inflammatory and proinflammatory actions. Another important aspect we are lacking is how some of the molecules are recognized by immune cells and which immune receptors are involved. Regarding the example of chitin in the *Aspergillus* cell wall, more information is needed regarding all the immune receptors that might recognize it, whether downstream signaling pathways might be “druggable”, and the possible utility of synthesis inhibitors. In addition, what is the role of TLR13 in *Aspergillus* recognition? Is this receptor’s function similar to that in the case of bacterial rRNA recognition? Another important area is to understand how *Aspergillus* glycosides contribute to the induction of immunity against aspergilli. How are other cell types, such as Th22, tissue-specific cells (airway epithelial cells), and their product cytokines and chemokines, involved in the immune response against aspergilli? Is there a specific immune cell type that plays a central role in regulating anti-*Aspergillus* immunity, or is there a multifaceted role for eliminating *Aspergillus* infection? There is still significant concern regarding single-nucleotide polymorphisms or mutations in specific immune molecules, such as PRRs. It is crucial to find genetic variations that are linked to a higher likelihood of infection or colonization by *Aspergillus* spp., or an increased immune response to them. Moreover, it is crucial to enhance our understanding of the interactions between *Aspergillus* spp. and our immune systems, as well as how immune responses will regulate and influence disease processes in different organs and tissues. An immune response to the invader is critical, but an overdone inflammatory response can be dangerous to the host. More research is needed to understand the control mechanisms that underlie this double-edged sword and attain the proper balance. The crosstalk of cytokines, chemokines, and immune cells in these diseases is illustrated in Figure 3.

To resolve these problems, it will be vital for clinicians, immunologists, and mycologists to collaborate in future endeavors. In patients with weakened immune systems, the equilibrium between proinflammatory and anti-inflammatory reactions may be disturbed. This imbalance can exacerbate the persistence and advancement of invasive aspergillosis. The role of immune cells, such as neutrophils and macrophages, circulating monocytes, dendritic cells, and natural killer cells, becomes critical in providing the optimal defense against *A. fumigatus* [177]. Thus, the rational development of therapies include how to boost the production of cytokines/chemokines from these immune cells during infection or reduce them when detrimental [178]. In order to modulate the protective immune response, IFN-γ has emerged as a promising immunotherapy against *Aspergillus* infection. IFN-γ therapy has been shown to enhance the antimicrobial activity of phagocytes and enhance survival; thus, IFN-γ holds promise as a cytokine-based adjunctive immunotherapy for invasive fungal infections, particularly in immunocompromised patients [148]. While antifungal drugs, such as voriconazole or amphotericin B, are frequently employed for the treatment of the illness, continuous research is being conducted in the field of modulating the immune response, including the use of immunomodulatory drugs. IFN-γ may also exert synergistic effects when used in combination with antifungal agents, such as amphotericin B or azoles [149]. This combination therapy has been explored in preclinical studies and has shown promise in enhancing the efficacy of antifungal drugs by augmenting host immune responses [148]. In addition, the *TNFR2* gene, which encodes tumor necrosis factor receptor type 2, plays a critical role in modulating the immune response, including inflammation and host defense against pathogens, such as *Aspergillus* species. Understanding the role of TNFR2 VNTR polymorphisms as genetic biomarkers of IPA susceptibility, and host ability to respond to infection by producing various cytokines, has clinical implications for prognostic risk stratification and personalized management strategies [147]. It is crucial to emphasize that the effective therapy of invasive aspergillosis necessitates a strong partnership between infectious disease specialists and other healthcare professionals. Treatment approaches may differ depending on the patient’s underlying health status. The timely detection and immediate commencement of suitable antifungal treatment are crucial for enhancing outcomes in patients with invasive aspergillosis.

## Figures and Tables

**Figure 1 jof-10-00251-f001:**
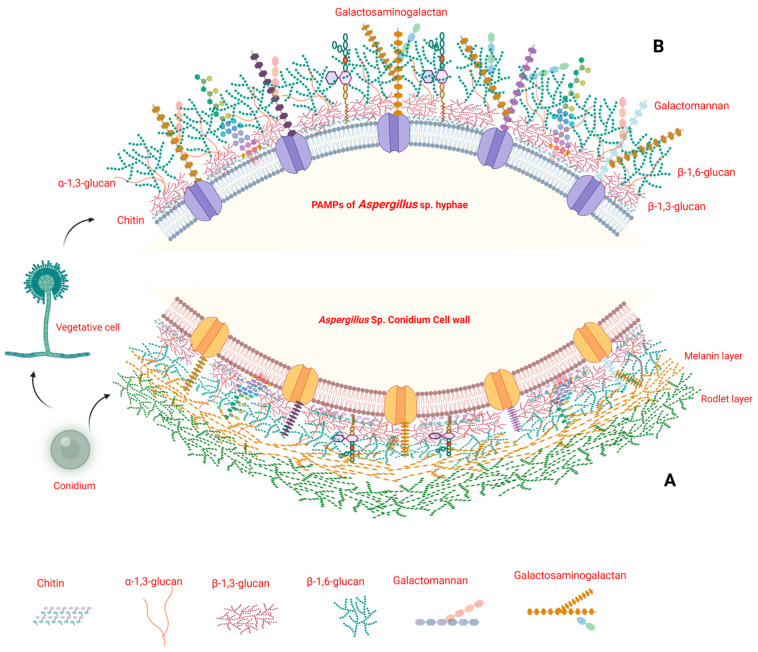
This figure represents the *Aspergillus* spp. conidium and hyphal cell wall components (PAMPs) of *Aspergillus* spp. (**A**) The conidial cell wall of *Aspergillus* spp., like other fungi, plays a critical role in protecting the fungal cell and interacting with the host environment. The cell wall of *Aspergillus* conidia is a complex structure consisting of various components, including the outer rodlet layer. These hydrophobic proteins contribute to the hydrophobicity of the conidial surface and may be involved in the interaction with host surfaces. Another component of the conidial layer is melanin. Melanin usually contributes to the protection of the aspergilli from environmental stress, and host immune responses. It plays a crucial role in the pathogenicity of *Aspergillus* spp. by shielding conidia from macrophages and epithelial cell phagocytic activity, hence preventing the acidification of phagolysosomes and phagocyte death. During germination, conidia shed their outer layer and thus allow cell wall components to interact with host immune cells via PRRs. (**B**) The cell wall of *Aspergillus* is a dynamic structure that undergoes remodeling during different stages of fungal growth and in response to environmental changes. The cell wall consists of complex carbohydrate components that are covalently attached to other components. Branched β-1,3-glucan/β-1,6-glucan is linked to chitin and galactomannan, whereas α-1,3-glucan and galactosaminogalactan fill the spaces between fibrillar polysaccharides. As germination starts, conidia lose the hydrophobic rodlet layer and the α-1,3-glucan moves from the inner layer to the outer layer of the conidium’s surface. Thereafter GAG appears on the cell wall surface and is involved in intrahyphal adhesion [34,35,36,38,41]. Created with Biorender.com.

**Figure 2 jof-10-00251-f002:**
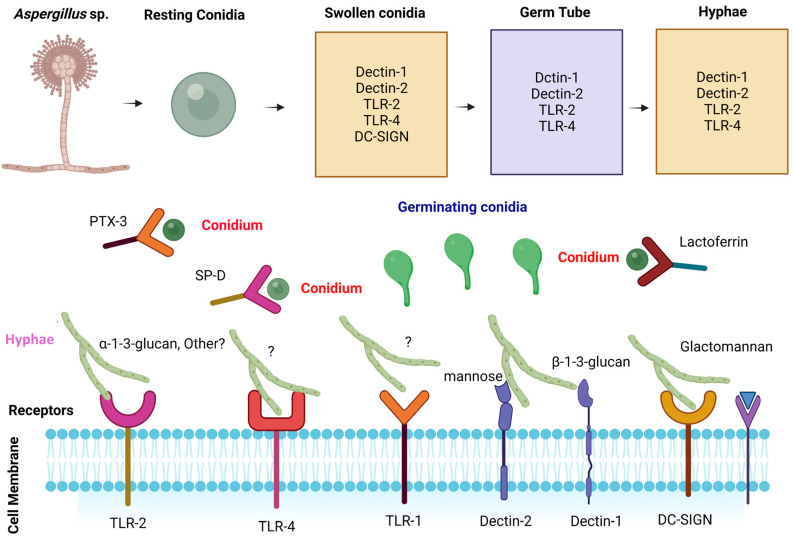
Recognition of *Aspergillus* spp. cell wall components (PAMPs) by pathogen recognition receptors (PRRs) of host cells. *Aspergillus* spp. are characterized by different morphotypes, a dormant stage or quiescent morphotype followed by the germinative type and hyphal type or mycelium type. Interactions of cell wall components in the aspergilli morphotypes with PPRs are documented. Aspergilli PAMPs are recognized by different pathogen recognition receptors present on immune cells, including C-type lectins (Dectin-1 and Dectin-2) and Toll-like receptors (TLR-1, 2, 3, and 4). Dectin-1 is involved in the recognition of β-glucan, whereas dectin-2 is involved in the recognition of aspergilli through the mannose moiety [43,59,60]. TLR-2 and TLR-4 are involved in the recognition of glucan residues of aspergilli [29]. TLR-9 and TLR-3 are endosomal receptors that recognize chitin and ds-RNA of *Aspergillus* spp. [61,62]. Soluble receptors, such as PTX3 [63], SP-D [64], and MBL [46], among others, assist in the recognition of *Aspergillus* conidia. A ‘?’ denotes that the PAMPs components are undefined. Created with Biorender.com.

**Figure 3 jof-10-00251-f003:**
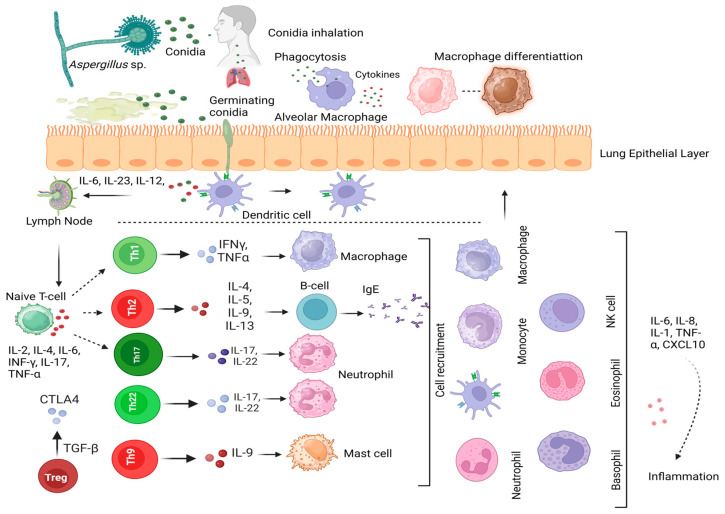
Interplay of host immune response during *Aspergillus* infections. Inhalation of *Aspergillus* spp. conidia leads to the development of different forms of aspergillosis: ABPA, CPA, and invasive aspergillosis. After the inhalation of conidia, they are initially encountered by innate immune cells, such as macrophages. Conidia and hyphae are recognized by PRRs present on immune cells. The interaction initiates engulfment by macrophages and the subsequent immune response [27,93]. Some of the conidia escape and start germination. Hyphae start to invade lung epithelial cells and are encountered by DCs via PRRs [21]. Subsequent interactions lead to the release of chemical messenger molecules, such as cytokine and chemokine responses, and DC migration to nearby lymph nodes, hence activating the adaptive immune response. The fungal structures interact with naïve T-cells that produce different cytokines and chemokines. These cytokines and chemokines serve as immune regulators to initiate the innate or adaptive response against aspergilli. Thus, the recruitment of immune cells, such as macrophages, neutrophils, eosinophils, etc., occur via cytokines and chemokines. They initiate the allergic or inflammatory response with dominant Th2 and Th9-type cell immunity with the help of IL-4, IL-5, IL-13, and IL-9 [113,115,116]. The immunoregulators act upon mast cells via IL-9, whereas, IL-4, IL-5, and IL-13 activate B-cells via Th2-type responses. This contributes to ABPA development. Alternatively, if IL-1, INF-γ, and TNF-α dominate the initial response against aspergilli exposure, they initiate Th1-cell-type immunity that leads to the clearance of *Aspergillus* infection. Further, IL-17 emergence leads to Th17-type cell immunity that also protects the host from aspergilli. A co-evolution of the Th1 and Th17 response has been observed against aspergilli [119,120,175,176]. This type of cell immunity has been observed in IA. Created with Biorender.com.

## Data Availability

Not applicable.

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
