# Peer review of "Interplay of Cytokines and Chemokines in Aspergillosis"

_jof, 2024, doi:10.3390/jof10040251_

Round 1

Reviewer 1 Report

The paper, entitled "Interplay of Cytokines and Chemokines in Aspergillosis", reviews the immune response mechanisms triggered by aspergillosis, a fungal infection caused by Aspergillus species. It discusses how Aspergillus pathogen-associated molecular patterns (PAMPs) are recognised by the host immune system, initiating a cascade of immune responses involving various cytokines and chemokines. 

The review is well written, and describes the dichotomy between Th1 and Th2 cell responses, highlighting their roles in the immune system's fight against aspergillosis.  Although this is a review article, the authors might consider reducing the length of the manuscript. 

Author Response

Reviewer 1

Major comments

The paper, entitled "Interplay of Cytokines and Chemokines in Aspergillosis", reviews the immune response mechanisms triggered by aspergillosis, a fungal infection caused by Aspergillus species. It discusses how Aspergillus pathogen-associated molecular patterns (PAMPs) are recognised by the host immune system, initiating a cascade of immune responses involving various cytokines and chemokines.

Detail comments

The review is well written, and describes the dichotomy between Th1 and Th2 cell responses, highlighting their roles in the immune system's fight against aspergillosis.

We thank the reviewer for the positive feedback on our review, particularly for recognizing our efforts to delineate the role of Th1 and Th2 cells in the immune system’s response against aspergillosis. 

Although this is a review article, the authors might consider reducing the length of the manuscript

A review article that aims to provide thorough and valuable insights into their subjects can get lengthy. Since the reviewer did not mention anything about the article that suggests which parts needs to be truncated, we are retaining what we view will be educational for the reader. However, overlapped content in the sections was consolidated in the revised version of the manuscript.

Reviewer 2 Report

This review by Shankar and colleagues provides a nice overview of the intricate role of cytokines and chemokines in various Aspergillus-induced diseases as well as the links between cytokine responses and the pathogen sensing system. I only have few comments to refine the article. In particular, section 3.2 would need some editing to improve its structure and conciseness, avoid redundancies, and highlight translational relevance.

Specific comments:

·         The first sentence of both the abstract and the introduction is suboptimal, as the review does not only capture invasive infection but also ABPA.

·         It would be nice to have a more visionary statement in the last sentence of the abstract, e.g., emphasizing a potential of cytokine patterns (e.g., in BAL) to guide biomarker-driven personalized management or even the deployment of cytokines or cytokine (receptor) antagonists as an immunotherapeutic strategy.

·         Line 61-62: I would suggest inserting the sentence later in the Introduction, after introducing the key principles of host defense. Further, the “immune circumvention” theme is not well described in section 2.1, e.g., with regards to the functions of the rodlet and melanin layer in evading or dampening innate effector responses. This is only described in the caption of Figure 1 but not in the main text.

·         While section 2.2 and Figure 2 address the “mainstream” PRRs, it would be nice to add a few sentences regarding emerging knowledge about other PRRs such as CD56 (NK cells) or EphA2 (epithelium).

·         Lines 248-259 can be deleted. All this has already been said in the Introduction.

·         I would suggest ending the introduction for section 3 after reference 73 (line 281) and discussing the following more granular information in the specific disease contexts in sections 3.1ff.

·         While section 3.1 is great and concise, section 3.2 seems unfocused and reiterates too much information about PRRs (lines 423-446). This part should be consolidated in section 2.2.

·         Lines 452-454: There are several syntax issues and missing words/prepositions.

·         The content of lines 447-451, lines 456 ff., and lines 467 ff. should be connected better and consolidated in a single, more concise paragraph. Currently, this information is repetitive and poorly organized, with unrelated information sprinkled in (e.g., discussion of unrelated chemokines in lines 454-456).

·         Throughout section 3.2, I am missing statements regarding translational relevance (diagnostic implications, prognostic risk stratification, immunotherapy). This is done very well in section 3.1 but not in section 3.2, although there is much more abundant literature for IA than for ABPA. Although immunotherapy is not the main focus of this review, there have been several studies for various cytokines suggesting an immunotherapeutic potential. These studies should be cited and the immunotherapeutic potential highlighted more clearly. I am particularly missing such discussion for INF-gamma, which is treated rather superficial in this review.

·        Why does the content of section 3.3 need to be in a separate chapter? This content could be integrated easily in section 3.2.

·         Figure 3 should not be in section 3.3 (or a consolidated section 3.2) but fits better in the concluding section 4, as it also refers to ABPA. Thus, it summarizes the content of both sections 3.1 and 3.2.

·         Some translational statements about the use of cytokines for prognostic risk stratification and cytokine-based immunotherapy would nbe warranted in section 4 (can be incorporated in lines 635ff.).

·         There are a couple of formatting issues (different fonts, missing spaces), but I assume these will be mitigated during typesetting.

Author Response

Reviewer 2

Major comments

This review by Shankar and colleagues provides a nice overview of the intricate role of cytokines and chemokines in various Aspergillus-induced diseases as well as the links between cytokine responses and the pathogen sensing system. I only have few comments to refine the article. In particular, section 3.2 would need some editing to improve its structure and conciseness, avoid redundancies, and highlight translational relevance.

Thank you for your positive feedback on our review article.

Detail comments

Specific comments:

The first sentence of both the abstract and the introduction is suboptimal, as the review does not only capture invasive infection but also ABPA.

The first line of introduction has been modified. 

Page 1.  The Aspergillus genus contains ⁓250 recognized species [1], of which 40 of them cause infection ranging from allergic to invasive aspergillosis [2].

It would be nice to have a more visionary statement in the last sentence of the abstract, e.g., emphasizing a potential of cytokine patterns (e.g., in BAL) to guide biomarker-driven personalized management or even the deployment of cytokines or cytokine (receptor) antagonists as an immunotherapeutic strategy.

Answer: We have modified the last sentence of the Abstract:

Page 1. Insight into the host response from both human and animal studies may aid in understanding the immune response in aspergillosis, possibly leading to harnessing the power of cytokines or cytokine (receptor) antagonists and transforming them into precise immunotherapeutic strategies. This could advance personalized medicine.

Line 61-62: I would suggest inserting the sentence later in the Introduction, after introducing the key principles of host defense.

Answer:  Line 61-62 has been moved to a later part of the introduction. Page 2 .

.

Further, the “immune circumvention” theme is not well described in section 2.1, e.g., with regards to the functions of the rodlet and melanin layer in evading or dampening innate effector responses. This is only described in the caption of Figure 1 but not in the main text.

Answer. Thank you. The functions of the rodlet and melanin layer in evading or dampening innate effector responses has been added. Page 3. With the onset of germination, a conidium loses its hydrophobic rodlet and melanin layer, and the previously hidden cell wall polysaccharides are revealed. The rodlet layer, formed of hydrophobins, contributes to immune evasion by encouraging adherence of conidia and biofilm formation, allowing the persistence of conidia within the host [38-39]. Melanin of conidia is involved in scavenging reactive oxygen species (ROS) and limiting the generation of nitric oxide by phagocytic cells [40]. Thus, the rodlet and melanin layer of conidia help in escaping or weakening the innate effectors response. The polysaccharide component, α-1-3-glucan moves from the inner layer to the outer layer of the surface.

While section 2.2 and Figure 2 address the “mainstream” PRRs, it would be nice to add a few sentences regarding emerging knowledge about other PRRs such as CD56 (NK cells) or EphA2 (epithelium).

Answer: We have added about CD56 in section 2.2. Page 6. CD56, another PRR [65], is primarily expressed on natural killer (NK) cells, and interacts with the cell wall component, GAG, of A. fumigatus. This interaction leads to the activation of NK cells, degranulation, and the production of effector molecules such as IFN-γ and TNF-α [66].

Answer: We have added about EphA2 in section 2.2. Page 7. E-cadherin on type II pneumocytes can mediate internalization of A. fumigatus conidia [68]. Another PRR, Ephrin type-A receptor 2 (EphA2) is present on oral and respiratory epithelial cells, and it binds to β-glucans during remodeling of the conidial cell wall [69]. EphA2 signaling has been implicated in phagocytosis by alveolar macrophages and epithelial cells, thereby promoting clearance of conidia, and the presence of DHN melanin on the conidia induces EphA2 dependent internalization of A. fumigatus conidia [70]. In addition, EphA2 activation stimulates the production of pro-inflammatory cytokines and chemokines, contributing to the recruitment and activation of other immune cells to the site of infection [70]

Lines 248-259 can be deleted. All this has already been said in the Introduction.

Answer: We consolidated it to make a flow of the content. 

I would suggest ending the introduction for section 3 after reference 73 (line 281) and discussing the following more granular information in the specific disease contexts in sections 3.1ff.

Answer: We suitably modified the manuscript, as suggested we moved the sentence from line 281 onward into the section 3.1. Page 9.

While section 3.1 is great and concise, section 3.2 seems unfocused and reiterates too much information about PRRs (lines 423-446). This part should be consolidated in section 2.2.

Answer: We suitably modified the manuscript, as suggested we moved the sentence information about PRRs (lines 423-446) to the section 2.2.Page 5. Dectin-2 is another member of the CLR family and is involved in the recognition of mannose-rich molecules and galactomannan on the Aspergillus cell wall [43].

Lines 452-454: There are several syntax issues and missing words/prepositions.

Answer: We corrected the errors in the revised manuscript.

The content of lines 447-451, lines 456 ff., and lines 467 ff. should be connected better and consolidated in a single, more concise paragraph. Currently, this information is repetitive and poorly organized, with unrelated information sprinkled in (e.g., discussion of unrelated chemokines in lines 454-456).

Answer: We have suitably modified the content (lines 447-467) highlighted in the revised manuscript. Page 12 Recruitment cytokines are essential for facilitating the migration of particular leukocytes to the location of infection in invasive aspergillosis. The cytokines that are crucial for protective responses against the infection are linked with Th-1 cells such as IL-12, IL-18, and interferon-gamma (IFN-γ). In contrast, the cytokines IL-4 and IL-10 of the Th2 cells have a role in the advancement of the infection. A study in immunocompromised mice observed a significant increase in TNF-α levels in the lung when exposed to Aspergillus conidia. Within this group of recruitment cytokines, the subset of CXC chemokines and their receptor CXCR2 plays a crucial role. These chemokines help in attracting neutrophils, whereas chemokines CCL3 and CCL2 are essential for attracting monocyte-lineage leukocytes and NK cells. Neutralizing TNF-α in these mice leads to a decrease in their ability to eliminate the fungal infection and an increase in mortality rates [93]. The inhalation of A. fumigatus conidia in mice leads to the production of IL-12, IL-18, and IFN-γ in the bronchoalveolar lavage fluid and the lung tissue [154]. In mice, when these cytokines are removed using neutralizing antibodies, the clearance of A. fumigatus from the lungs is delayed [154]. It has been observed that the depletion of TNF-α led to reduced activity of myeloperoxidase in the lungs and a decrease in the synthesis of chemokines that attract inflammatory cells to the lung.

Throughout section 3.2, I am missing statements regarding translational relevance (diagnostic implications, prognostic risk stratification, immunotherapy). This is done very well in section 3.1 but not in section 3.2, although there is much more abundant literature for IA than for ABPA. Although immunotherapy is not the main focus of this review, there have been several studies for various cytokines suggesting an immunotherapeutic potential. These studies should be cited and the immunotherapeutic potential highlighted more clearly. I am particularly missing such discussion for INF-gamma, which is treated rather superficial in this review.

Answer: We suitably modified the manuscript; discussion on INF-γ has been added. However, the main focus of the articles was not immunotherapy against aspergillosis. Page 12. Immunotherapy, such as with IFN-γ, shows potential, particularly in patients with impaired immune systems [145]. This occurs through the activation of macrophages that boosts their antifungal activity [146]. IFN-γ promotes Th1 cell development and increases the production of antifungal cytokines, including IL-12 and TNF-α [146-147]. Exogenous treatment of IFN-γ can enhance the host's immune response against Aspergillus, leading to better outcomes in some individuals in some clinical research studies [148]. IFN-γ may synergize with conventional antifungal therapy [149].

TNF-α, like IFN-γ, activates PMNs, leading to increased oxygen radical release and hyphal injury to A. fumigatus in vitro studies [143,150]. Apart from these cytokines, colony stimulating factors (G-CSF, GM-CSF and M-CSF) have been reported to enhance the capacity of the innate immune response against mycoses, including invasive aspergillosis ([151-153].

 Why does the content of section 3.3 need to be in a separate chapter? This content could be integrated easily in section 3.2.

Answer: We integrated the section 3.3 into the 3.2. Page 14 . Recent studies focused on fungal extracellular vehicles (EVs) have shown their role….

   Figure 3 should not be in section 3.3 (or a consolidated section 3.2) but fits better in the concluding section 4, as it also refers to ABPA. Thus, it summarizes the content of both sections 3.1 and 3.2.

Answer: Thank you. We moved figure 3 in the concluding section 4. Page 15.

Some translational statements about the use of cytokines for prognostic risk stratification and cytokine-based immunotherapy would be warranted in section 4 (can be incorporated in lines 635ff.).

Answer: We added translational the following translational statement. Page 16. The role of immune cells such as neutrophils and macrophages, circulating monocytes, dendritic cells, and natural killer cells become critical in providing the optimal defense against A. fumigatus [177]. Thus rational development of therapies include how to boost the production of cytokines/chemokines from these immune cells during infection or reduce them when detrimental [178]. In order to modulate the protective immune response, IFN-γ has emerged as a promising immunotherapy against Aspergillus infection. IFN-γ therapy has been shown to enhance the antimicrobial activity of phagocytes, and enhanced survival, thus, IFN-γ holds promise as cytokine based adjunctive immunotherapy for invasive fungal infections, particularly in immunocompromised patients [148]. Whereas antifungal drugs, such as voriconazole or amphotericin B, are frequently employed for the treatment of the illness, continuous research is being conducted in the field of modulating the immune response, including the use of immunomodulatory drugs. IFN-γ may also exert synergistic effects when used in combination with antifungal agents such as amphotericin B or azoles [149]. This combination therapy has been explored in preclinical studies and has shown promise in enhancing the efficacy of antifungal drugs by augmenting host immune responses [148]. In addition, the TNFR2 gene, which encodes the tumor necrosis factor receptor type 2, plays a critical role in modulating the immune response, including inflammation and host defense against pathogens such as Aspergillus species. Understanding the role of TNFR2 VNTR polymorphisms as genetic biomarkers of IPA susceptibility, and host ability to respond to infection by producing various cytokines, has clinical implications for prognostic risk stratification, and personalized management strategies [147].

There are a couple of formatting issues (different fonts, missing spaces), but I assume these will be mitigated during typesetting.

Answer: We corrected the errors in the revised manuscript.

Reviewer 3 Report

Figures need to be revised, some sentences are not complete/correct, and there should be spell-check.

This review article is about the interplay of cytokines and chemokines during Aspergillosis.

Comments:

1.     Lines 90-91: Here, the authors are dealing with host-Aspergillus interaction. Therefore, I suggest changing the subtitle as “Role of Pathogen-Associated Molecular Patterns and Pathogen Receptors during host-Aspergillus interaction.

2.     Line 94: What the authors mean “small”?

3.     Line 127: during germination not only rodlet layer, but conidia also lose their melanin layer.

4.     Figure 1 need to be checked: (i) Linear beta-1,6-glucan is represented; provide reference for the presence of linear beta-1,6-glucan in the Aspergillus cell wall; (ii) Where is alpha-1,3-glucan in the vegetative cell; (iii) GAG is a heteropolysaccharide, while a linear homogenous monomeric units have been presented; (iv) It is galactomannan, and not glactomannan; and it is not linear; (v) some of the polysaccharide structures are not indicated with names.

5.     CLRs and TLRs are mentioned, but what about NLRs and RLRs? Do they play any role against Aspergillus species?

6.     Authors should differentiate immune cell bound and soluble PRRs.

7.     References are incomplete in Figure 2, for e.g., PTX3, SP-D and what about MBL mentioned in the text?

8.     Lines 228-229: statement is not correct, as, chitin recognition has been reported. Later on, the authors themselves provide information about chitin receptors. What about chitin recognition by Ig-opsonization and through immunoglobulin receptors?

9.     What about complement receptors and Aspergillus recognition?

10.  Section 3.2: The authors should introduce better how there will be cytokine response during invasive aspergillosis by suppressed immune system.

11.  Figure 3: Authors can show differentiated macrophages in different colors.

12.  Line 635: how immune balance against Aspergillus is maintained?

Author Response

Reviewer 3

Major comments

Figures need to be revised, some sentences are not complete/correct, and there should be spell-check.

Answer: Thank you. We corrected the errors in the revised manuscript. Figures 1 & 3 have been revised.

Page 4 Figure 1, & Page 15 Figure 3

Detail comments

This review article is about the interplay of cytokines and chemokines during Aspergillosis.

 Comments:

 Lines 90-91: Here, the authors are dealing with host-Aspergillus interaction. Therefore, I suggest changing the subtitle as “Role of Pathogen-Associated Molecular Patterns and Pathogen Receptors during host-Aspergillus interaction.

Answer: We suitably modified the manuscript. Page 2. Role of Pathogen-Associated Molecular Patterns and Pathogen Recognition Receptors during host-Aspergillus interaction

Line 94: What the authors mean “small”?

Answer: We suitably modified the manuscript. Page 2. Replaced ‘small conserved molecular patterns’ with ‘small molecules with conserved motif’

Line 127: during germination not only rodlet layer, but conidia also lose their melanin layer.

Answer: We suitably modified the manuscript, following content has been added. Page 3. With the onset of germination, a conidium loses its hydrophobic rodlet and melanin layer, and the previously hidden cell wall polysaccharides are revealed. The rodlet layer, formed of hydrophobins, contributes to immune evasion by encouraging adherence of conidia and biofilm formation, allowing the persistence of conidia within the host [38-39]. Melanin of conidia is involved in scavenging reactive oxygen species (ROS) and limiting the generation of nitric oxide by phagocytic cells [40]. Thus, the rodlet and melanin layer of conidia help in escaping or weakening the innate effectors response. The polysaccharide component, α-1-3-glucan moves from the inner layer to the outer layer of the surface.

Figure 1 need to be checked: (i) Linear beta-1,6-glucan is represented; provide reference for the presence of linear beta-1,6-glucan in the Aspergillus cell wall; (ii) Where is alpha-1,3-glucan in the vegetative cell; (iii) GAG is a heteropolysaccharide, while a linear homogenous monomeric units have been presented; (iv) It is galactomannan, and not glactomannan; and it is not linear; (v) some of the polysaccharide structures are not indicated with names.

Answer: We modified the figure 1.

(i) Branched beta-1,6-glucan has been represented  in the updated figure 1.

(ii) alpha-1,3-glucan in the vegetative cell has been added

(iii) GAG, as a heteropolysaccharide, has been updated

(iv) Corrected

(v) polysaccharide structures are now indicated with names

Page 4

CLRs and TLRs are mentioned, but what about NLRs and RLRs? Do they play any role against Aspergillus species?

Answer: We suitably modified the manuscript. Page 7. Apart from surface bound and endosomal PRRs, cytosolic PRRs such as NOD like receptor (NLR) are also stimulated by PAMPs [77]. NLR family includes nucleotide binding oligomerization domain 1(NOD1), NOD2, NLRP3/Cryopyrin/Nalp3 ‘inflammasome’ and NLRC4/Ipaf inflammasome [78-79]. Study on human monocyte cell lines discovered that Aspergillus hyphal fragments induce NLRP3 inflammasome assembly and IL-1β cytokine production. Further, study showed that A. fumigatus conidia increase NOD2 expression in alveolar epithelial cells and macrophages as well as in lungs of mice in pulmonary invasive aspergillosis model [80].

Moreover, retinoic acid-inducible gene-1 encodes (RiG-1)-like receptor (RLRs), another intracellular PRR that recognizes PAMPs. RLRs include RIG-1, melanoma differentiation-associated 5 (MDA5) and laboratory genetics and physiology 2 (LGP2). Study on A. fumigatus demonstrated that MDA5/MAVS (mitochondrial antiviral signaling) signaling is essential to resist pulmonary A. fumigatus infection in a murine model. MDA5/MAVS activates in response to ds-RNA of live A. fumigatus and induces type III IFN expression and production of the CXCL10 chemokine. It has also been demonstrated that neutrophil killing of Aspergillus conidia depends upon MDA5/MAVS signaling [81].

Another study demonstrated that genetic polymorphisms in MAVS alter the production of chemokines, creating a risk for the patient for invasive pulmonary aspergillosis. In a mouse model of A. fumigatus infection, it has also been demonstrated that alveolar macrophages are the key cells where the MDA5/MAVS dependent-interferon response is induced [82].

Authors should differentiate immune cell bound and soluble PRRs.

Answer: We suitably modified the manuscript re PRRs, Page 4. PRRs are categorized into soluble and cell surface bound receptors. Soluble receptors such as collectins (that include MBL, SP and CL-11) as well as PTX-3 and ficolins, are reported to be important for the recognition of Aspergillus spp. Apart from these, cell surface bound receptors such as Toll-like receptors (TLRs) and dectins also play significant roles in the recognition of aspergilli and initiate immune response against them [42].

References are incomplete in Figure 2, for e.g., PTX3, SP-D and what about MBL mentioned in the text?

Answer: We suitably modified sentence with references. Page 6 line 244. Soluble receptors such as PTX3 [63] SP-D [64] and MBL [46] among others assist in recognition of Aspergillus conidia.

Lines 228-229: statement is not correct, as, chitin recognition has been reported. Later on, the authors themselves provide information about chitin receptors. What about chitin recognition by Ig-opsonization and through immunoglobulin receptors?

Answer: We suitably modified the manuscript.  Page 7. Beckman et al observed the chitin induced an anti-inflammatory signature characterized by the production of IL-1Ra in the presence of human serum, production which was abrogated in immunoglobulin-depleted serum. Fcγ-receptor-dependent recognition and phagocytosis of IgG-opsonized chitin was identified as a novel IL-1Ra-inducing mechanism [49].

What about complement receptors and Aspergillus recognition?

Answer: We suitably modified the manuscript, following are lines added. Page 5. Resting conidia trigger the alternative complement pathway. As conidia start to swell and are transformed to hyphae, the classical pathway dominates [44]. Study showed that the MBL of the lectin complement pathway binds to carbohydrate moieties on the Aspergillus cell wall and activates complement pathways by increasing the deposition of the C4 component of complement pathway. Furthermore, MBL can support C3 cleavage by a C2 bypass mechanism after contact with A. fumigatus conidia, resulting in activation of the alternative pathway and avoiding formation of the classical pathway C3 convertase [45].

Section 3.2: The authors should introduce better how there will be cytokine response during invasive aspergillosis by suppressed immune system.

Answer: We suitably modified the manuscript; following addition has been made in the section 3.2.

Page 11. Immunotherapy, such as with IFN-γ, shows potential, particularly in patients with impaired immune systems [145]. This occurs through the activation of macrophages that boosts their antifungal activity [146]. IFN-γ promotes Th1 cell development and increases the production of antifungal cytokines, including IL-12 and TNF-α [146-147]. Exogenous treatment of IFN-γ can enhance the host's immune response against Aspergillus, leading to better outcomes in some individuals in some clinical research studies [148]. IFN-γ may synergize with conventional antifungal therapy [149].

TNF-α, like IFN-γ, activates PMNs, leading to increased oxygen radical release and hyphal injury to A. fumigatus in vitro studies [143,150]. Apart from these cytokines, colony stimulating factors (G-CSF, GM-CSF and M-CSF) have been reported to enhance the capacity of the innate immune response against mycoses, including invasive aspergillosis ([151-153].

  1. Figure 3: Authors can show differentiated macrophages in different colors.

Answer: We modified figure 3. Page 15.

  1. Line 635: how immune balance against Aspergillus is maintained?

Answer: We suitably modified the manuscript, following lines are added to demonstrate how to boost the immune response and maintain immune balance against Aspergillus infection.

Page 16. An immune response to the invader is critical, but an overdone inflammatory response can be dangerous to the host.  More research is needed to understand the control mechanisms that underlie this double-edged sword, and attain the proper balance

The role of immune cells such as neutrophils and macrophages, circulating monocytes, dendritic cells, and natural killer cells become critical in providing the optimal defense against A. fumigatus [177]. Thus rational development of therapies include how to boost the production of cytokines/chemokines from these immune cells during infection or reduce them when detrimental [178]. In order to modulate the protective immune response, IFN-γ has emerged as a promising immunotherapy against Aspergillus infection. IFN-γ therapy has been shown to enhance the antimicrobial activity of phagocytes, and enhanced survival, thus, IFN-γ holds promise as cytokine based adjunctive immunotherapy for invasive fungal infections, particularly in immunocompromised patients [148]. Whereas antifungal drugs, such as voriconazole or amphotericin B, are frequently employed for the treatment of the illness, continuous research is being conducted in the field of modulating the immune response, including the use of immunomodulatory drugs. IFN-γ may also exert synergistic effects when used in combination with antifungal agents such as amphotericin B or azoles [149]. This combination therapy has been explored in preclinical studies and has shown promise in enhancing the efficacy of antifungal drugs by augmenting host immune responses [148]. In addition, the TNFR2 gene, which encodes the tumor necrosis factor receptor type 2, plays a critical role in modulating the immune response, including inflammation and host defense against pathogens such as Aspergillus species. Understanding the role of TNFR2 VNTR polymorphisms as genetic biomarkers of IPA susceptibility, and host ability to respond to infection by producing various cytokines, has clinical implications for prognostic risk stratification, and personalized management strategies [147].

Round 2

Reviewer 3 Report

The authors have made relevant modifications.

I am fine with the modificantion made in the revised manuscript.